# The Application of Ferric Chloride-Lignin Sulfonate as Shale Inhibitor in Water-Based Drilling Fluid

**DOI:** 10.3390/molecules24234331

**Published:** 2019-11-27

**Authors:** Rongjun Zhang, Long Gao, Wenguang Duan, Weimin Hu, Weichao Du, Xuefan Gu, Jie Zhang, Gang Chen

**Affiliations:** 1Shaanxi Province Key Laboratory of Environmental Pollution Control and Reservoir Protection Technology of Oilfields, Xi’an Shiyou University, Xi’an 710065, China; rongjunzhang@xsyu.edu.cn (R.Z.); gaolong22330@163.com (L.G.); cg1014@126.com (W.H.); duweichao@xsyu.edu.cn (W.D.); xuefangu@xsyu.edu.cn (X.G.); 2CNPC Xibu Drilling Engineering Company Limited, Urumqi, Xinjiang 830026, China; duanwg_zj20@cnpc.com.cn; 3State Key Laboratory of Petroleum Pollution Control, CNPC Research Institute of Safety and Environmental Technology, Beijing 102206, China

**Keywords:** ferric chloride, lignin sulfonate, inhibitor, shale, swelling

## Abstract

A series of ferric chloride-lignin sulfonate (FCLS) was prepared from ferric chloride and lignin sulfonate to be used as shale inhibitor. The swelling rate of clay with FCLS-2 (*w/w* = 0.3%) decreased to 41.9%. Compared with control, FCLS-2 displayed high inhibitive ability against the hydrating and swelling processes of clay. Thus, the swelling degree of samples with FCLS-2 was much lower than that of the control, as well as the mud ball was more stable in FCLS-2 solution. Essentially, these excellent performances in inhibitor were assigned to the hydrogen bonding, electrostatic interaction and anchoring between FCLS-2 and other components. In addition, FCLS-2 has good compatibility with other common drilling fluid additives, and it can reduce the viscosity of systems, regardless of the room temperature or high temperature.

## 1. Introduction

Shale oil/gas has been one of the topic of interests in the world in recent years. Drilling is the first and most important process in the shale oil/gas exploration. Water-based drilling fluids are considered economical and environmentally friendly compared to synthetic and oil-based drilling fluids in the shale oil/gas exploration. Quite different from the normal reservoir, the stability of borehole suffers some problems mostly because of hydrating and swelling of water sensitive shale during the drilling processes, including borehole wash-out, stuck pipe, disintegration of cuttings, and bit balling [1,2,3]. Theoretically, when the water sensitive shale (high montmorillonite content) is immersed in water-based drilling fluid, the swelling or dispersing behavior of shale will occur rapidly, which relates to the chemical composition of shale and the drilling fluid. Thus, various shale inhibitors have been utilized widely in drilling operations, including polymeric, amine-based, ionic liquids, and surfactant-based shale inhibitors [4,5]. However, most traditional additives face some challenge because of their deficiency in the kindly environmental requirements [6], and the processing of waste after drilling is relative difficult and high cost, especially for the oil-based shale inhibitors [7]. Recently, natural organic chemicals have drawn much attention as shale inhibitor in crude oil extraction due to their advantages, such as friendly environmental instinct, high inhibition ability, stable rheological performance, and high lubrication [8,9,10]. Currently, natural products have been developed as shale inhibitor in many water-based oilfield working fluids owing to their superior compatibility. In this work, the inhibitive performance as well as probable mechanism of ferric chloride-lignin sulfonate agent were investigated by using mud balls immersing test, linear expansion experiments, particle distribution measurements, thermogravimetric analysis (TGA), and scanning electron microscopy image (SEM).

## 2. Results and Discussion

### 2.1. Optimization of Reaction Conditions

The effect of the mass ratio of ferric chloride and lignin sulfonate on the inhibitive performance was evaluated via the clay-swelling rate as shown in Table 1. Among the six formula, the minimal swelling rate of clay was obtained when the mass ratio of ferric chloride and lignin sulfonate was 4:1. Thus, FCLS-2 agent was selected in the following experiments. 

### 2.2. Test of Swelling Inhibition

The dynamic inhibiting performance of aforementioned FCLS against the swelling process of bentonite was measured via the linear swelling rate. As shown in Figure 1, the swelling rate of control increased dramatically during the first 10 min, and then increased slightly during the further 50 min. While, the swelling rates of systems with inhibitors decreased significantly compared with control and reached to a minimum in 0.3% FCLS-2 samples. It was noticed that the swelling rate does not agree with a linear relation with different concentration of FCLS-2. This phenomenon revealed the complex inhibition mechanism ascribed to the adsorption of ferric chloride-lignin sulfonate on the clay surface through co-interaction, including electrostatic interaction, hydrophobic interaction and hydrogen bonds based on an anchoring effect. Under this co-interaction, swelling rate of samples with FCLS-2 (*w/w* = 0.3%) dropped from 69.0% to 41.9%.

In addition, the mud ball test was employed to visually observe the effect of FCLS-2 on clay. As shown in Figure 2, the mud ball immersed in tap water evidently swelled within 12 h, and its surface became completely rough and loose after 24 h. While, the mud ball immersed in 0.3% FCLS-2 solution changed slightly in 12 h, 24 h, or 36 h, and its surface was still smooth except for a little crack after 36 h (Figure 3). The remarkable inhibition of FCLS-2 on the swelling process of clay mostly attribute to the absorption of FCLS-2 on the surface of clay, which blocked or reduced the water penetration into the clay, and prevent the clay from hydrating swelling.

### 2.3. Performance in Water-Based Drilling Fluids

The performance of FCLS-2 in water-based drilling fluids was measured in accordance with GB/T 16783.1-2006, and the results are shown in Table 2. 

With nearly the same low fluid loss performance, the viscosity and shearing force of the drilling fluid added FCLS-2 became lower than the control sample without inhibitor, but the filtration property does not present any significant change, indicating that FCLS-2 can restrain the bentonite slurry deep swelling to reduce the viscosity effectively. The results also indicated that FCLS-2 is compatible with the common drilling fluid additives, modified starch, and PAM, and it does not enhance the filtration as added in the composed water-based drilling fluid. The most important thing is that FCLS-2 can reduce the viscosity under high temperature, which can be used as a viscosity reducer. 

### 2.4. Particle Size Distribution Test 

The effect of FCLS-2 on bentonite particle size distribution was also studied since size distribution are closely related to reveal the swelling process. Figure 4 shows that the average size of original bentonite particles was 38 μm, and the hydrated samples reduced to 8 μm (Table 3). When the LS or FCLS-2 was added in the hydrated bentonite respectively 16 h later, the average size of both the systems increased to some extent compared with the water-treated samples, especially the particle size of FCLS-2 treated samples. The obvious larger particle size in samples with FCLS-2 was the consequence of the flocculation function of FCLS-2.

### 2.5. TGA 

The ratio of mass loss and temperature is a crucial factor to evaluate the inhibitor [11]. The temperature-dependent weight loss of different modified bentonite samples was measured using TGA technology as shown in Figure 5. The weight loss-temperature curves of FCLS-2 or LS-treated bentonites significantly differs from the control from 50 °C to 180 °C, particularly the former. The weight loss of tap water-treated bentonite was up to 3.2% from 50 °C to 180 °C, while that of the 0.3% FCLS-2 modified bentonite was down to 0.32%. This obvious difference indicated that the FCLS-2 can reduce the water amount of the sample except that the organic compounds will decompose above 180 °C. The result also revealed that FCLS-2-treated bentonites possess better thermal stability than LS-treated samples in the temperature above.

### 2.6. SEM

In order to further study the inhibition mechanism of FCLS-2, SEM was used to observe the morphology of bentonite with different modification. SEM images displayed that the particle size of 0.3% FCLS-2-immersed bentonite was much larger than tap water-treated bentonite after 24 h as shown in Figure 6. The result was consistent with their distribution of clay particle size determined by laser particle size analyzer. On the whole, the performance of FCLS-2 not only prevent bentonite from hydrating and swelling, but also pull and gather most of the hydrated and swollen clay [12,13].

## 3. Materials and Methods 

### 3.1. Materials and Reagents 

Sodium lignin sulfonate and ferric chloride were purchased from Xi’an Chemical Reagent Co. Ltd. (Xi’an, China). Bentonite, modified-starch, and PAM were all supplied from Yanchan Oilfield chemical company (Shaanxi, China). 

### 3.2. Preparation of FCLS

Sodium lignin sulfonate (LS) and ferric chloride (FC) were dissolved in distilled water respectively with the concentration of 20%. Then ferric chloride solution was added into the lignin sulfonate solution with different mass ratio dropwise under stirring at room temperature [14,15]. As shown in Figure 7, the final product, ferric chloride-lignin sulfonate (FCLS) solution was achieved after stirring for 1 h at 60 r/min.

### 3.3. Evaluation of Inhibitive Ability

The swelling behavior of bentonite in water was evaluated using a shale expansion instrument (NP-01, Haitongda, Co., Ltd., Chuangmeng, Qingdao, China), according to Chinese Petroleum and Natural Gas Industry Standards SY/T5971-1994 and SY/T6335-1997. Mud ball immersed experiment was conducted as follows: 10 g of bentonite and 10 mL of tap water were used to form a mud ball, and then it was placed in 80 mL of aqueous solutions or tap water for 24 h [16,17]. The appearances of the mud balls were recorded and compared. The general error of the data is ±0.2%.

### 3.4. Performance in Drilling Fluid 

The water based drilling fluid was prepared with a dosage of 4% bentonite (*m/m*), and the apparent viscosity (AV), plastic viscosity (PV), yield point (YP), API Filtration (FL), and friction coefficient (tg) was evaluated using a viscometer (ZNN-D6S, Haitongda, Co., Ltd., Qingdao, China) according to the reported methods [18,19,20]. The general error of the data is ±1%.

### 3.5. Particle Distribution Measurement

The bentonite was dispersed in a certain solution or tap water with a dosage of 4% (*w/w*), and was stirred for 24 h. After inhibitor was added to the system, it was stirred for another 20 min. Then the particles size distribution was measured by a laser particle size analyzer (LS-13320, Beckman Coulter, Inc., Brea, CA, USA) under the reported method [14,15]. The general error of the data is ±2%.

### 3.6. TGA and SEM

The bentonite samples were dispersed in tap water or system with inhibitor over 24 h. Then, the bentonite was separated from the system and dried at 105 °C for TGA and SEM image. TGA experiment was conducted on a TGA/DSC thermal analysis instrument (1/1600, METTLER TOLEDO, Inc., Columbus, OH, USA) at a ramp of 20 °C/min from room temperature to 825 °C under nitrogen flow. The surface morphology of the bentonite samples was evaluated by a digital microscope imaging scanning electron microscope (model SU6600, serial No. HI-2102-0003) at a 40.0 kV accelerating voltage on the basis of the reported method [14,15].

## 4. Conclusions

In this work, ferric chloride-lignin sulfonate (FCLS) was synthesized with ferric chloride and lignin sulfonate and their performance as shale inhibitor in water-based drilling fluid was systematically investigated. The result demonstrated that FCLS-2 displays effectively inhibitive ability against the hydrating and swelling processes of clay. The swelling rate of 0.3% FCLS-2 treated clay decreased from 69.0% to 41.9%. In addition, the mud ball added FCLS-2 was more stable, and its swelling degree was much lower compared with control. Furthermore, the inhibition mechanism of FCLS-2 to shales was explored using SEM combined with TGA and particle size analyzer. The probable mechanism is that FCLS-2 not only prevents bentonite from hydrating and swelling via hydrogen bonding, ion exchange, and anchoring effect, but also pulls and gathers most of the hydrated and swollen clay.

## Figures and Tables

**Figure 1 molecules-24-04331-f001:**
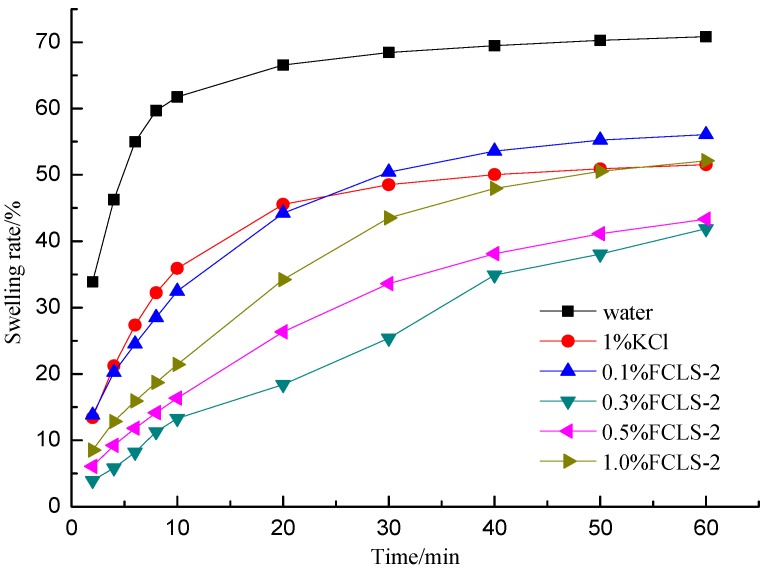
The effect of the inhibitors concentration.

**Figure 2 molecules-24-04331-f002:**
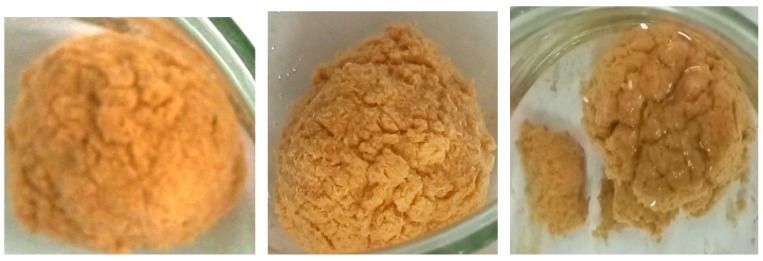
The appearance of mud balls immersed in water for 12 h, 24 h, and 36 h.

**Figure 3 molecules-24-04331-f003:**
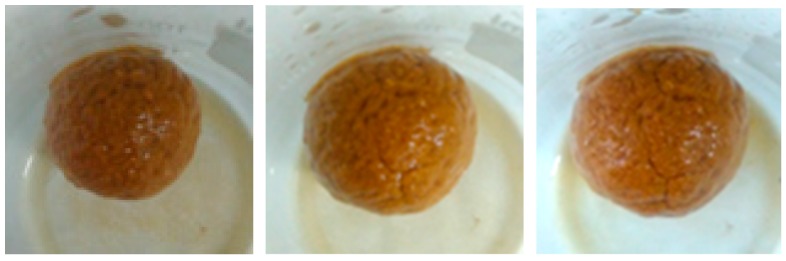
The appearance of mud balls immersed in FCLS-2 for 12 h, 24 h, and 36 h.

**Figure 4 molecules-24-04331-f004:**
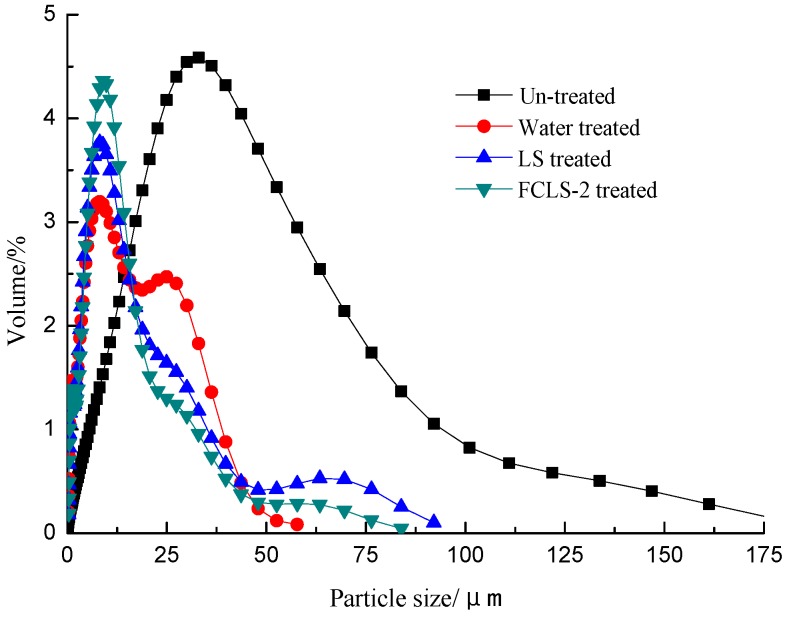
The distribution of clay particle size in different suspensions.

**Figure 5 molecules-24-04331-f005:**
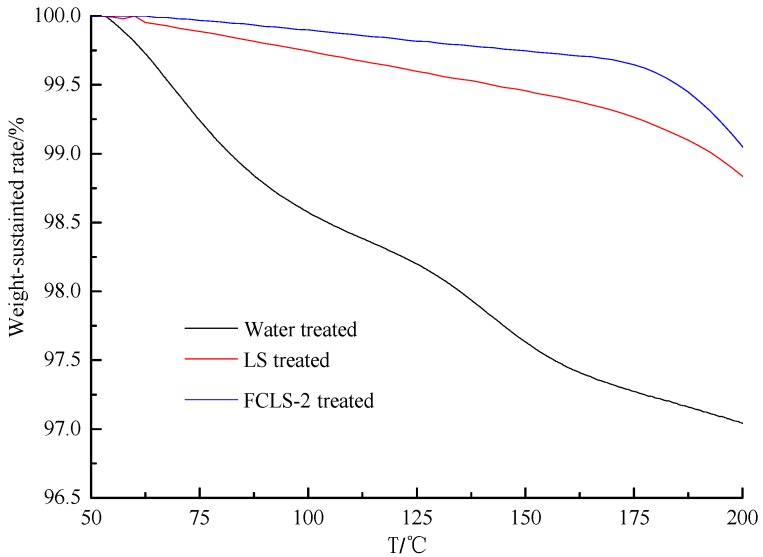
The thermogravimetric analysis (TGA) of different bentonite samples.

**Figure 6 molecules-24-04331-f006:**
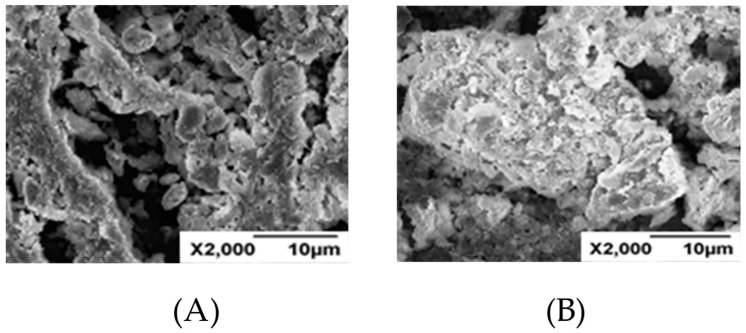
Scanning electron microscopy image (SEM) images of bentonites: (**A**) hydrated bentonite; (**B**) bentonite treated with FCLS-2.

**Figure 7 molecules-24-04331-f007:**
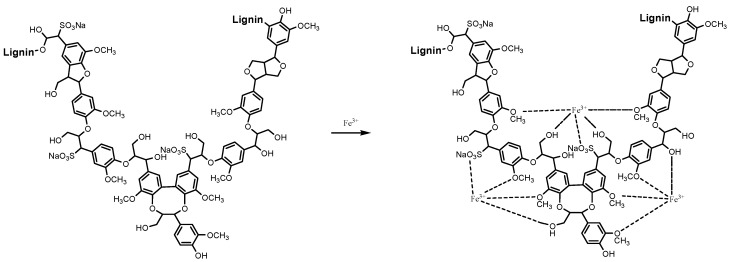
Synthetic principle of ferric chloride-lignin sulfonate inhibitor (FCLS).

**Table 1 molecules-24-04331-t001:** The conditions of synthesis ferric chloride-lignin sulfonate inhibitors.

Name	Mass Ratio	Swelling Rate (%) (60 Min)
LS	/	52
FC	/	65
FCLS-1	2:1	54
FCLS-2	4:1	42
FCLS-3	6:1	44
FCLS-4	8:1	44

**Table 2 molecules-24-04331-t002:** Evaluation results of drilling fluid rheological properties.

Additives	T/°C	AV	PV	YP	YP/PV	FL	tg
		/(mPa·s)	/(mPa·s)	/Pa	Pa/(mPa·s)	/mL	
Blank	25	3.6	2.1	1.40	0.67	12.0	0.07
	120	2.6	1.8	0.70	0.39	14.0	0.18
PAM	25	8.0	4.2	3.88	0.92	13.0	0.03
	120	7.8	4.8	3.30	0.74	15.7	0.03
PAM +0.3%FCLS-3	25	3.6	3.1	0.51	0.16	16.2	0.03
	120	6.0	4.5	1.20	0.25	15.2	0.04
Modified starch	25	4.9	2.8	2.15	0.77	11.0	0.09
	120	9.8	8.5	2.50	0.15	15.7	0.11
Modified starch +0.3%FCLS-3	25	3.5	2.2	1.33	0.60	10.1	0.19
	120	5.7	4.5	1.20	0.55	15.5	0.11

**Table 3 molecules-24-04331-t003:** Particle size of bentonite treated with different methods.

Additives (180 °C)	Mean/μm	Median/μm
Un-treated	38	27
Water treated	8	5
0.3%LS	8	6
0.3%FCLS-2	10	8

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
