# Peer review of "The Application of Ferric Chloride-Lignin Sulfonate as Shale Inhibitor in Water-Based Drilling Fluid"

_molecules, 2019, doi:10.3390/molecules24234331_

Round 1

Reviewer 1 Report

Page 1, Line 4; The word “and” is in a wrong place. Either use it between the last two authors or relocate the “Weichao Du and Xuefan Gu” at the end.

Page 2-8, There is no consistency between numbers and units. Please make sure to follow the journal instructions. Either leave a space or none. For example, Page 4, lines 98-100; there is no space between 12h, and yet on page 7, line 146; there is a space between 24 h. This is also true for other units i.e., 25°C.

Page 2, Line 46; The procedure for preparation of FCLS is not adequate. Needs more detail.          

Page 2, Fig. 1; The author does not indicate the type of LS used in this investigation at all. However, the figure 1 shows Na-LS. This is rather confusing. In my opinion, it is imperative that the nature of LS should be mentioned since the preparation of FCLS may vary between mono- (Sodium, Ammonium) and divalent (Calcium and Magnesium) LS.   

Page3, Line 82; Change the “experiment” to “experiments”

Page 8; Please re-examine the format of the cited journals.

Author Response

Page 1, Line 4; The word “and” is in a wrong place. Either use it between the last two authors or relocate the “Weichao Du and Xuefan Gu” at the end.

We have revised it.

Page 2-8, There is no consistency between numbers and units. Please make sure to follow the journal instructions. Either leave a space or none. For example, Page 4, lines 98-100; there is no space between 12h, and yet on page 7, line 146; there is a space between 24 h. This is also true for other units i.e., 25°C.

We have revised the space between numbers and units.

Page 2, Line 46; The procedure for preparation of FCLS is not adequate. Needs more detail.         

The detail was added.

Page 2, Fig. 1; The author does not indicate the type of LS used in this investigation at all. However, the figure 1 shows Na-LS. This is rather confusing. In my opinion, it is imperative that the nature of LS should be mentioned since the preparation of FCLS may vary between mono- (Sodium, Ammonium) and divalent (Calcium and Magnesium) LS.  

Lignin sulfonate is a by-product in the papermaking. It is separated from cellulose by using SO2 and NaOH or Ca(OH)2, so there are at least two lignin sulfonate products. In this work we used sodium lignin sulfonate.

Page3, Line 82; Change the “experiment” to “experiments”

We have revised it.

Page 8; Please re-examine the format of the cited journals.

The references of format were double cheeked and revised.

Reviewer 2 Report

In this work author synthesized a series of ferric chloride-lignin sulfonate used as shale inhibitor. The authors designed experiments in a systematic way. The results may merit to publish in Molecules. The abstract is meaningful and clearly describes a comprehensive summary of the research. The authors studied the mechanism and proposed using SEM and TGA. I fully agreed with their proposed mechanism. I have  only concern about the introduction part. The introduction part is not well written. The motivation should be clear along with the state of the art in this field. The author may refer to some good review like Fuel 251 (2019): 187-217. The manuscript lack discussion in results and discussion section. The language quality and editing of the manuscripts need attention as well.

Author Response

We have reorganized the "Introduction part", and the language was double-checked.